# Autonomic, endocrine, and psychological stress responses to different forms of blood draw

**Tierney K. Lorenz**[ORCID]*

Department of Psychology and Center for Brain, Biology and Behavior, University of Nebraska-Lincoln, Lincoln, NE, United States of America

* tierney.lorenz@unl.edu

## Abstract

### Background

Although fingerstick is often favorably compared to venipuncture as a less invasive method of drawing blood for clinical labs, there is little empirical research that compares physical and psychological stress responses to fingerstick vs. venipuncture (blood draw using a needle in the arm) within the same person.

### Methods and findings

We assessed changes in cortisol (a stress hormone), heart rate variability (a marker of autonomic stress), and psychological stress in 40 healthy women who completed both venipuncture and fingerstick. Contrary to our predictions, there was a significant decline in cortisol across conditions, with greater decline from pre- to post-draw in response to venipuncture than fingerstick. There were similar patterns of rise and fall in heart rate variability in both types of blood draw, suggestive of mild vasovagal responses. Psychological measures of stress (such as negative emotion and perceived stress) were generally stronger predictors of participant's reported pain and blood draw preferences than physical stress measures.

### Conclusions

These findings challenge the characterization of fingerstick as necessarily "less invasive" than venipuncture, as participant's stress responses to fingerstick were equivalent to (and for some measures greater than) their response to venipuncture. Heart rate variability response to fingerstick significantly predicted that individual's vasovagal-like responses to venipuncture, suggesting that measuring heart rate variability during pre-donation hemoglobin testing may identify donors at risk for adverse events during venipuncture.

**Data Availability Statement:** Deidentified data and code to replicate these analyses are available at https://osf.io/7xc3k/.

**Funding:** This work was supported by internal funds from the UNL Department of Psychology, the Nebraska Tobacco Settlement Biomedical

Research Development Fund, and the UNL Office of Research and Economic Development, awarded to TKL. The funders had no role in study design, data collection and analysis, decision to publish, or preparation of the manuscript.

**Competing interests:** The authors have declared that no competing interests exist.

## Introduction

When studying stress response, it is important to understand the ways in which biomarker collection may be stressful in and of itself. Much research has investigated predictors of psychological stress during venipuncture, and to a lesser extent, capillary draw from the fingertip (aka fingerstick); however, to date, no study has compared results within the same individual and thus a true comparison across methods cannot be made. Taking the view that venipuncture is itself stressful, health researchers often turn to fingerstick, which is reported to be less invasive than venipuncture and assumed to be less stressful (see [1, 2]). However, this assumption has not been well tested. In the present study, we assessed changes in cortisol, heart rate variability, and self-reported stress to venipuncture and fingerstick blood draw in healthy volunteers, and tested psychophysiological predictors of participant self-reported pain and preferences for future blood draw.

Among healthy participants undergoing venipuncture, there is on average an increased physical stress response immediately prior to draw, followed by significant *decreases*; however, there is considerable individual variation in the magnitude of this effect [3]. This pattern appears to hold for indices of both short-term, autonomic stress response such as heart rate (HR) and heart rate variability (HRV) and longer-term, hypothalamic pituitary adrenal (HPA) axis stress response such as cortisol [3, 4]. These findings are consistent with research suggesting that not only pain, but the *anticipation* of pain, are potent cues to elicit stress responses [5–7]. Consistent with this interpretation, HRV analyses suggest that the decreases in HR in the five minutes following venipuncture are due specifically to increased parasympathetic activity–that is, increased recovery responses [8]. What is more, changes in both autonomic and HPA stress markers occur even with repeated exposures [3, 9]. In short, venipuncture can introduce changes in stress response that could interact with other experimental procedures in ways that make interpretation of experimental manipulations of stress quite difficult.

One proposed solution to this concern is using less invasive blood collection methods, such as fingerstick. Fingerstick offers many valuable features: it can be conducted by people with minimal training in phlebotomy including the participant themselves, raising the possibility of out-of-lab studies [10]. Fingerstick can be used to generate dried blood spots, which are stable with minimal refrigeration and thus well-suited for field research, epidemiologic studies and bio-banking [11]. Indeed, fingerstick has been a primary method for several landmark studies of psychosocial determinants of health such as the Add Health study [12, 13] and the NSHAP study [14]. There are also a variety of point-of-care instruments that can use a drop of blood from fingerstick to generate almost immediate testing results that can be shared with the participant [15]. Finally, because it takes considerably less skill to find a participant's fingertip than to locate a vein, fingerstick methods may produce lower rates of missing data.

Because fingerstick methods require smaller needles and less blood than venipuncture, they are sometimes considered "minimally invasive" and assumed to have minimal effects on stress [16–19]. However, testing this assumption has led to mixed results. Most clinical research on fingerstick stress has been conducted in children, as fingerstick (and related heelstick) are more commonly used in pediatric care. In infants and children, fingerstick and/or heelstick has been shown to reliably induce an HPA response [20], especially when combined with other typical clinical procedures (e.g., immunization injection [21]). However, these analyses suggest that children's stress response diminish with increasing age [20].

Research on fingerstick-related stress in adults is considerably sparser. To date there has been no empirical study on the effect of fingerstick in and of itself on HPA response among healthy adults. Only two studies, both from the same primary investigator, report on the effects of fingerstick alone on autonomic indices of stress in adults. Both of these studies found no

significant effect of fingerstick on heart rate [17, 18]. However, as noted above there is ample evidence that anticipation of pain can induce stress in healthy adults; thus, the fact that the fingertip has considerably high density of pain receptors than typical venipuncture sites [22] suggests that the procedure may induce stress in people who anticipate pain. Indeed, research in people who use fingerstick for routine monitoring of chronic health conditions, or who are frequent blood donors, find that fingertip testing is commonly perceived as more painful than puncture at other sites and associated with higher self-reported psychological stress [23–25].

In sum, although there are many assumptions about the effect of blood draw on physical and psychological stress response in healthy adults, empirical data comparing different methods within the same individual are lacking. In the present study, we tested if fingerstick and venipuncture would elicit different responses in autonomic, endocrine, and psychological stress indices. We also explored if stress measured during each blood draw method predicted participant's preferences for that type of draw, perceived pain, and psychological stress.

## Material and methods

This project was preregistered on the Open Science Framework (https://bit.ly/2DZFPjm). All procedures were approved and overseen by the Institutional Review Board at the University of Nebraska-Lincoln. Participant recruitment ran from April to October of 2019.

### Participants

Forty-five women were recruited from the community using flyers and email announcements to University listserves, and from the Psychology participant pool. Participants all provided written informed consent and received their choice of either $20 or course credit for participating. A secondary purpose of the study was to assess interactions between women's immune response and their sexual history (see preregistration for details), and thus all participants were cis-gendered women with some lifetime history of partnered sexual activity. Exclusion criteria included use of medications likely to alter stress response and/or hormone measures (e.g. glucocorticoids), contraindications for salivary measures (e.g., recent dental work), contraindications for blood draw (e.g., weighs less than 110 lbs; no prior history of blood draw; history of prior complications with blood draw), and evidence of hormone dysfunction (e.g., significant menstrual disturbances). Of the 45 participants who enrolled, 40 completed both experimental sessions; however, analyses include all available data including that from dropouts. The final sample had an average age of 21.93 (SD = 4.00), and was predominantly White non-Hispanic (67%) with 17% Asian, 10% Latina/Hispanic, 3% Black, and 3% multiracial (17% of participants did not provide their race/ethnicity).

### Procedures

Participants completed two experimental sessions of one hour each, spaced 1–3 days apart. To account for diurnal variation in cortisol [26], all sessions were scheduled in the afternoon; moreover, participants provided data on time since waking, which was entered as a covariate in all analyses using cortisol data. In one lab session, participants underwent venipuncture and in the other, they underwent fingerstick; the order of these two procedures were randomized across participants. Participants were asked to avoid eating, drinking, or smoking in the hour prior to their experimental session.

Both sessions followed similar order of procedures. First, participants were given an 8oz (237mL) bottle of water to drink and swish around to clean their mouth. Although hydration status has been shown to impact blood pressure [27], these pressor effects are generally not seen in young healthy women until fluid ingestion is 500ml or greater [27–29].

Once they finished drinking, participants were hooked up to an electrocardiogram (ECG) which measured their heart rate continuously throughout the session. Three sites corresponding to Einthoven's triangle [30]; upper right and left shoulder just above the clavicle, and left ankle) were prepared with a baby wipe to remove any sweat or lotions, and three Ag/Al non-latex contact electrodes with conductive hydrogel (Medsource model MS-65005) were applied. Data were sampled at 2000 samples/sec, which is well above the minimum needed for accurate measurement of HRV [31].

For the duration of the session, participants were seated on a typical clinic exam table that was partially reclined; that is, as participants were neither fully supine nor upright, they were able to complete survey measures on a laptop while avoiding postural confounding effects on HRV [32]. All materials were accessible to the participant on a side table so they did not need to change posture. Once the participant was comfortable, the researcher left the room. Participants were given 2–3 minutes to come to rest (and to ensure at least 10 minutes had passed since they drank water), then completed an unstimulated passive drool sample of 2mL.

After completing the baseline saliva sample, participants alerted the researcher via intercom. The researcher re-entered the room to complete blood draw (see below), clean and bandage the participant, and then left the room. Participants rested and watched a series of pleasant nature scenes for 2–6 minutes following blood draw, then provided a post-draw saliva sample; the length of time watching nature scenes was manipulated to ensure all participants started their post-draw sample precisely 10 minutes after the needle/lancet was applied. They were then instructed via intercom to complete a series of surveys on a laptop. At 20 minutes following blood draw, participants completed a final post-recovery saliva sample. Following this, the researcher re-entered the room and unhooked the participant from the ECG. Immediately after the session, saliva samples were transported to a -80C freezer where they were kept until day of assay.

**Blood draw procedures.** In both draw types, we attempted to align methods as much as possible: for example, participants were in the same setting and physical position for both (partially reclined on the exam table), and had both draws from their non-dominant hand/arm; the researchers wore the same protective equipment (e.g., gloves, lab coat), and washed their hands before and after draw; participants got the same 3-2-1 countdown prior to insertion of the needle/lancet; and researchers were instructed to make the same distracting small talk with a standardized list of approved topics (e.g., the weather, the participant's hometown) while setting up equipment and completing the draw. Similarly, to ensure fidelity to these standardized methods, all researchers completed the same kinds of training for both methods, including three group trainings in biological safety and best practices, at three supervised practice draws followed by a checkout with the project PI, and at least one surprise inspection.

*Venipuncture.* On one day, participants provided a sample of 20mL whole blood via standard venipuncture procedures [33]. Specifically, a tourniquet was applied to the non-dominant arm, and the anterior cubital fossa (a large vein on the inside of the elbow) was located. Following confirmation of draw site, the tourniquet was released and the site was cleaned using isopropyl alcohol wipes. The researcher assembled the draw unit (a 21-gauge needle attached to a vacutainer holder) and reapplied the tourniquet immediately prior to draw. The researcher inserted the needle and applied two 10mL pressure-sealed vacutainers in rapid succession. As the second tube was filling, the tourniquet was released; after the second tube was completed, the needle was withdrawn, and the participant was given a sterile gauze pad to press into the draw site. The participant was asked to raise their arm up to slow bleeding and kept it up while the researcher cleaned up and labeled the tubes (approx. 1 minute). Finally, the participant lowered their arm and the researcher bandaged the site with a new piece of sterile gauze and a bandage. The total time for draw was approximately 5 minutes.

All venipuncture draws were completed by two researchers with training in phlebotomy and at least two years of experience with doing venipuncture draws in healthy research participants. If the first stick was unsuccessful in drawing sufficient volume, researchers completed a second stick; however, there were never more than two attempts. Less than 10% of participants had two sticks. It should be noted that the volume of blood taken was very small (less than 0.5% of a typical person's total blood volume), and too low to directly reduce total concentrations of circulating cortisol to a detectable degree.

*Fingerstick*. On the other day (counterbalanced for order effects), participants provided a sample of < 0.5mL whole blood via standard capillary blood draw methods [33]. Specifically, participants were given a warm compress to hold for 1 minute. The researcher took the compress and cleaned the ring finger of the non-dominant hand with isopropyl alcohol wipes, allowing the site to dry completely. The researcher then applied a 21-gauge (1.8mm) contact-activated lancet to the fingertip, at a site midway down the fingertip (parallel to the nailbed) and adjacent to the finger pad. The first drop of blood was wiped away with a sterile gauze pad, and 5–7 subsequent drops were applied to two filter paper cards. If there was poor flow, the researcher would gently massage the hand to stimulate flow and/or re-wipe the site with another sterile gauze pad. After both filter paper cards were completed, the participant was given a gauze pad to press into the draw site. As with venipuncture, the participant was asked to raise their hand to slow bleeding, after which the researcher applied a sterile bandage. The total time for draw was approximately 5–8 minutes, reflecting higher variability in blood flow from finger capillaries relative to veins [34].

Fingerstick was completed by the same researchers who did venipuncture, as well as four additional research assistants who were given extensive training in capillary blood draw (see above). Although we allowed for one potential re-stick (as with venipuncture), there was no participant for whom fingerstick did not produce sufficient volume, so there were no re-sticks.

## Measures

**Measures of subjective stress to blood draw.** Each study day, shortly after blood draw, participants completed several measures of subjective stress. They indicated their perceived levels of psychological and physical stress using a slider anchored from 0 (no stress) to 100 (very stressed). They completed a brief version of the Positive and Negative Affective Schedule (PANAS [35]), which assessed their current emotional state. They also were asked to rate how typical that day's blood draw procedures were, relative to other times they had their blood drawn (e.g., at the doctor's office). On the second study day only, participants were asked to indicate which kind of blood draw was more painful, which was more stressful, and which they would prefer in the future if they had to have blood drawn.

**Physical stress measures.** *Cortisol*. Saliva samples were assayed for cortisol via well-validated enzyme-linked immunosorbent assay (ELISA) kits, following manufacturers instructions (Salimetrics, LLC, Carlsbad, CA). There were 4 samples (1.6% of total samples) that could not be assayed due to insufficient sample volume or evidence of blood contamination; these data were considered missing at random. Samples were run in duplicate; inter- and intra-assays coefficients of variance (CVs) confirmed good assay consistency (7.8% and 7.7%, respectively). Inspection of the raw data revealed non-normality due to a few extremely high values, which is common in salivary cortisol data [36]. Thus, we winsorized the raw data using R's *psych* package [37], setting the threshold to correct no more than 0.5% of the total dataset; this method of dealing with cortisol outliers has been shown to reduce error in final parameter estimates [38]. However, to test the robustness of effects, we re-ran all analyses with and without outliers, and report on any differences between these models.

*Heart Rate (HR) and Heart Rate Variability (HRV).* As noted above, HR was sampled continuously before, during and after blood draw procedures using an ECG (Biopac Systems Inc, Goleta, CA). We binned these data into discrete 5-minute blocks reflecting baseline (quiet rest prior to the researcher entering the room for the draw procedures), during the draw, immediately post-draw, and recovery (starting 20 minutes post-draw). Beat-to-beat RR intervals were collected using the AcqKnowledge peak finder function. These RR intervals were submitted to the Kubios HRV analysis program (Kubios Oy, Kuopio, Finland). We used Kubios' automatic artifact correction algorithm to identify and remove movement artifacts and ectopic beats [39]; any file for which there were more than 10% of total beats identified as artifacts was flagged for recleaning.

The HRV measure of interest was the standard deviation of the RR interval (RRSD), the most widely used measure owing to its simple calculation and robustness against artifacts [40]. The RRSD is thought to reflect autonomic balance: higher values broadly indicate greater parasympathetic dominance, relative to the contributions of the sympathetic nervous system; but see also Billman [41] for a more nuanced perspective. As the RRSD can be biased by changes in overall heart rate [42–44], we included HR as a covariate in all analyses of HRV.

We had additionally proposed in our pre-registration to additionally use the ratio of the low-frequency and high-frequency bands of a Fourier transformation of the HRV signal (LF/HF ratio), as this measure was thought to more accurately assess autonomic balance in response to stress. However, since then we have been made aware of several studies that have called this assumption into serious question, particularly in female participants. As such, as such we were convinced to drop this element of the analytic plan.

*Other Measures.* On each study day, participants completed a brief survey assessing issues related to salivary analysis including time since waking. As there are substantial diurnal variations in cortisol secretion tied to sleep-wake patterns [45], time since waking was entered as a covariate for all analyses including cortisol data. Participants also provided demographic information such as age and race/ethnicity. As there are significant effects of age on both cortisol and HRV [46, 47], age was a covariate for all analyses including physiological data.

## Results

Complete data, code and output for all models described below is available on the OSF project page (https://bit.ly/2ZLmoDt).

### Manipulation check

A majority of participants (57%) reported the lab blood draws as "about the same" to other times they had their blood drawn (e.g., at the doctor's office), with no differences between venipuncture and fingerstick ($\chi^2(2) = 0.25$, $p = 0.96$; Table 1).

**Table 1. Participant ratings of how stressful each type of draw was, relative to a typical draw (e.g., at the doctor's office).**

| | Less stressful than typical blood draw | | About the same as a typical blood draw | | More stressful than a typical blood draw | |
|---|---|---|---|---|---|---|
| | n | % | n | % | n | % |
| Venipuncture | 12 | 30% | 24 | 60% | 4 | 10% |
| Fingerstick | 15 | 34% | 24 | 55% | 5 | 11% |

These ratings were taken each day.

## Changes in autonomic and endocrine markers of stress during fingerstick and venipuncture

For these analyses, we conducted repeated measures linear mixed models, specifying a random intercept by participant (nested within draw type) to account for individual differences at baseline. Although there was no evidence of significant order effects for any parameter, we controlled for order of the two lab sessions as this was part of our preregistered analysis plan. As noted above, we also controlled for age in both cortisol and HRV analyses, as well as time since waking for the cortisol analyses.

**Endocrine stress markers.** There was a significant main effect of time on change in cortisol ($F(2, 148.86) = 6.31$, $p < 0.01$), with a marginally significant interaction between draw type and time ($F(2, 148.86) = 2.92$, $p = 0.06$; Fig 1). Specific contrast testing revealed a significant difference between venipuncture and fingerstick at the post-draw timepoint (*Effect estimate* = -0.0420, $t = -2.39$, $p = 0.02$), such that cortisol levels dropped significantly from pre- to post-draw in the venipuncture condition but not in fingerstick condition. Including outliers did not change the direction or significance of these effects.

**Autonomic stress markers.** The main effect of time on change in HRV was significant ($F(3, 206.30) = 29.70$, $p < 0.01$), but the interaction between draw type and time was not ($F(3, 206.30) = 1.08$, $p = 0.36$). Across both draw types, there was a significant increase in HRV from pre-draw to the moment of blood draw, followed by a return to baseline at post-draw (Fig 2). Post-hoc testing of the robustness of the effect confirmed that dropping heart rate as a covariate did not alter the direction or significance of these effects.

To further probe these effects, we conducted an exploratory (not pre-registered) analysis, re-running the model with heart rate as the outcome. There was a significant interaction

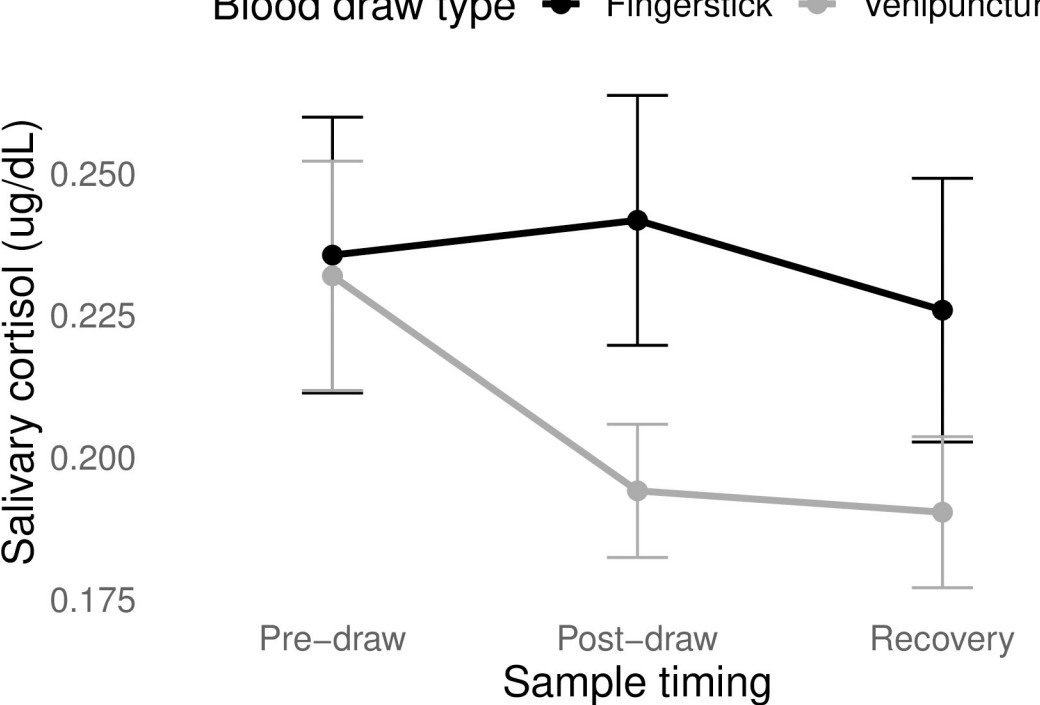

**Fig 1. Changes in salivary cortisol in response to different types of blood draw.** Error bars represent the standard error of the mean. There was a significant difference between draw types at the post-draw timepoint.

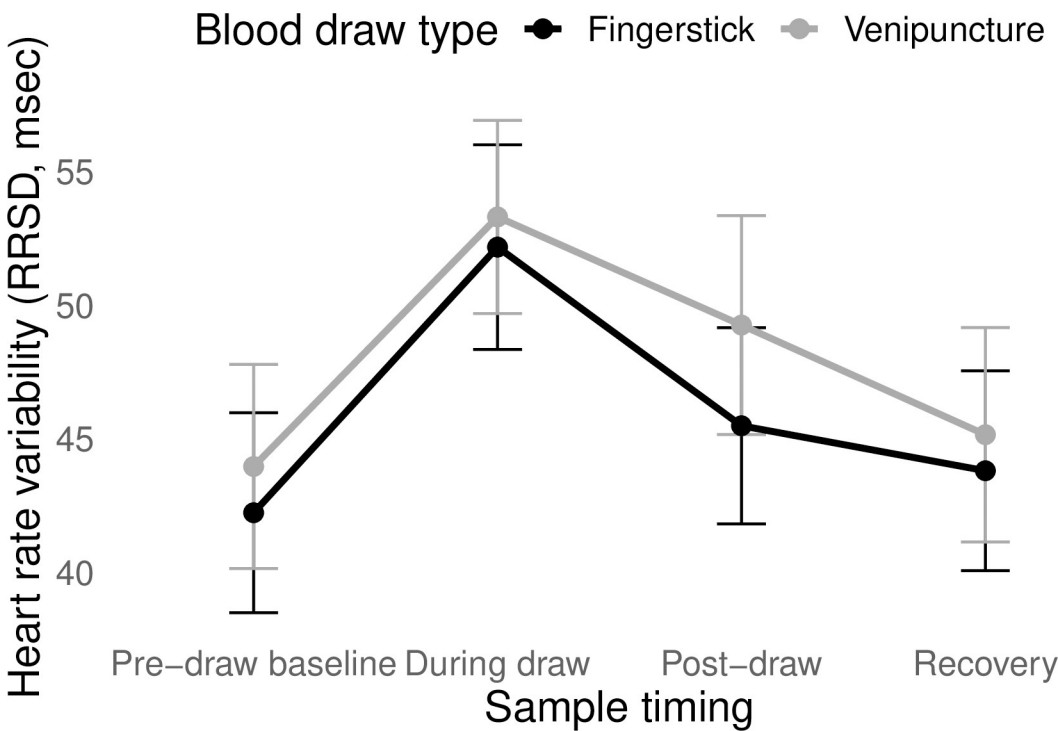

**Fig 2. Changes in heart rate variability in response to different types of blood draw.** The measure of heart rate variability was the standard deviation of successive inter-beat intervals (RRSD). Error bars represent the standard error of the mean. The change in HRV over time did not differ significantly by draw type.

between draw type and time ($F(3, 204) = 4.99$, $p < 0.01$), such that fingerstick was associated with a sharper decline in HR from pre-draw to the moment of blood draw and faster return to baseline (Fig 3).

**Exploring non-linear change in cortisol and Heart Rate Variability (HRV).** Plotting the cortisol and HRV data over time suggested that changes in stress may be non-linear. Thus, as an exploratory post-hoc test, we re-ran the above models including a quadratic effect of time, comparing model fit with a pseudo-$R^2$ using the method outlined in Nakagawa, et al. [48]. These analyses confirmed that in both cases, the non-linear effect of time was significant (cortisol: $F(2, 148.86) = 6.31$, $p < 0.01$; HRV: $F(2, 212.07) = 30.1$, $p < 0.01$). However, fit for linear vs. non-linear models was similar across both models of cortisol (linear pseudo-$R^2$: 0.096; non-linear pseudo-$R^2$: 0.096), and HRV (linear pseudo-$R^2$: 0.477; non-linear pseudo-$R^2$: 0.498), suggesting the main effects are adequately captured in the linear models.

### Psychological stress and emotion following venipuncture and fingerstick

**Same-day psychological measures.** As above, we conducted repeated measures mixed linear models to assess differences between draws in psychological stress measures; however, as self-report measures were only collected post-draw, the random intercepts by participant were not nested (i.e., there was only one value per day, and thus nesting within day was not necessary).

**Emotion following blood draw.** There were no significant differences between draw types in post-draw emotion, for either negative emotion ($F(1, 41.79) = 0.10$, $p = 0.75$) or positive emotion ($F(1, 41.47) = 0.29$, $p = 0.59$).

**Fig 3. Changes in heart rate in response to different types of blood draw.** Error bars represent the standard error of the mean. The change in heart rate over time did not differ significantly by draw type.

**Self-reported physical and psychological stress.** There were no differences between draw types in participants perceived stress, for either perceived physical ($F(1, 38.60) = 0.17$, $p = 0.68$) or psychological stress ($F(1, 38.51) = 0.13$, $p = 0.72$).

### Psychological stress as predictors of autonomic and endocrine stress response

To test interactions of within-session physical and psychological stress reactivity, we conducted repeated measures linear mixed models similar to those described above, but included level of self-reported physical stress and psychological stress in the model as an interaction term.

Change in cortisol was predicted by the interaction between draw type and self-reported physical stress ($F(2, 822.95) = 6.75$, $p < 0.01$) as well as the interaction between draw type and psychological stress ($F(2, 822.98) = 3.75$, $p = 0.02$). In each case, the general pattern was the same: for venipuncture, lower perceived stress was associated with a greater decline in cortisol from pre- to post-draw. For fingerstick, however, higher perceived stress was associated with a significant *rise* in cortisol from pre- to post-draw, while lower perceived stress was associated with a moderate (non-significant) decline in post-draw cortisol (Fig 4).

Change in HRV was significantly predicted by the interaction between draw type and self-reported physical stress ($F(3, 738.38) = 8.65$, $p < 0.01$). In venipuncture sessions, lower perceived physical stress was associated with lower HRV overall and faster return to baseline following blood draw, while for fingerstick sessions, perceived physical stress did not impact change in HRV (Fig 4). Change in HRV was not predicted by psychological stress ($F(3, 739.42) = 1.80$, $p = 0.15$).

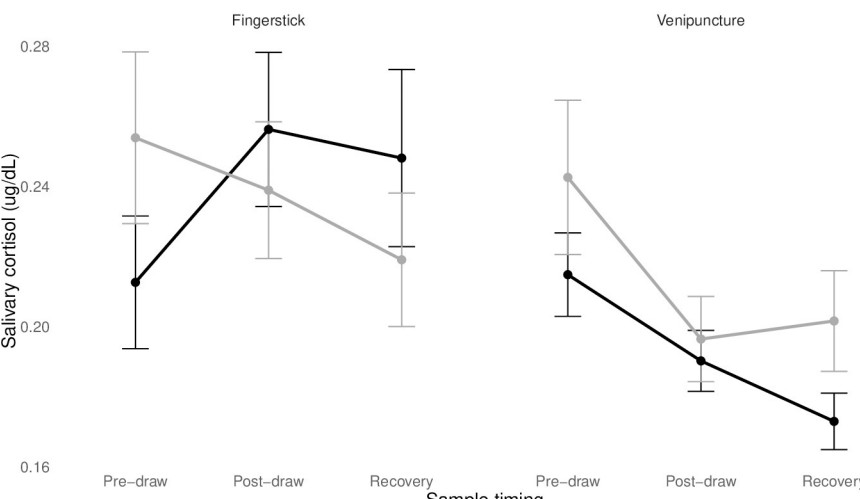

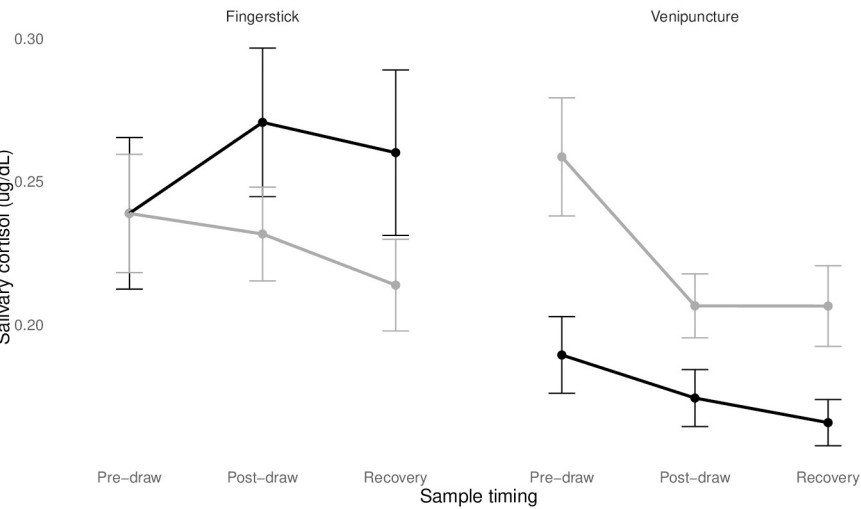

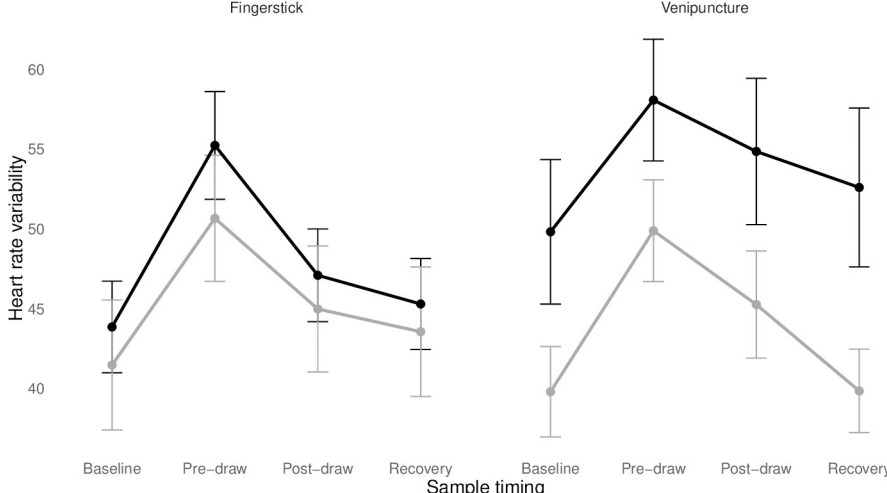

**Fig 4. Interactions between blood draw type, changes in cortisol and HRV, and self-reported physical and psychological stress following blood draw.** Self-reported stress was split at the mean for display purposes. The measure of HRV was the standard deviation of successive inter-beat intervals (RRSD); higher values indicate greater parasympathetic and/or less sympathetic nervous system activity. Error bars represent the standard error of the mean.

## Preferences and comparisons of stress and pain

After participants had completed both types of blood draw, they were asked to rate which day had been more stressful, which more painful, and which type of draw they would prefer in the future. We used simple $\chi^2$ tests of independence to test for differences between groups. Significantly more participants thought venipuncture was the more stressful draw ($\chi^2(2) = 8.45$, p = 0.02; Table 2). However, participants were relatively evenly split on which draw type they thought was more painful ($\chi^2(2) = 0.80$, p = 0.67). Participants were significantly less likely to state they had "no preference" than would be expected by chance ($\chi^2(2) = 6.20$, p = 0.05), but were relatively evenly split on whether they would prefer venipuncture or fingerstick. In other words, although participants expressed definite preferences, these preferences were not overwhelmingly in favor of one type or the other.

## Psychological vs. physiological stress measures as predictors of participant preferences for blood draw

Finally, we examined which measure of stress would best predict which draw type participants rated as more painful and stressful, and which they would prefer in the future. For these analyses, we collapsed the cortisol and HRV data across time using a transformation of area under the curve with respect to ground [49], which summarizes each factor as the integral of its rise and fall over time. We then conducted stepwise multinomial logistic regressions with choice as the outcome variable, and the area under the curve for HRV and cortisol, self-reported stress and emotion, the effect of condition (venipuncture vs. fingerstick), and their interaction, as possible predictors. The stepwise function found the best model fit among all combinations of predictors according to the Akaike Information Criterion.

**Predictors of which draw type was more stressful.** In the best fit model for predicting which draw type was rated as more stressful, the only predictor was positive emotion: the more positive emotion participants reported (following either kind of blood draw), the less likely they were to state they thought both types were equally stressful (Fig 5).

**Predictors of which draw type was more painful.** In the best fit model for predicting which draw type was rated as more painful, there were three predictors: change in HRV, self-reported psychological stress, and negative emotion. The greater the change in HRV, the less likely participants were to rate both types of draw as equally painful. The greater a participants' psychological stress following either kind of blood draw, the more likely they were to rate fingerstick as more painful draw. And the more negative emotion they reported following either blood draw, they more likely they were to name venipuncture as more painful (Fig 6).

**Table 2. Participant ratings of stress, pain, and preferences by draw type.**

| | Which was more stressful? | | Which was more painful? | | Which would you prefer in the future? | |
|---|---|---|---|---|---|---|
| | **n** | **%** | **n** | **%** | **n** | **%** |
| Venipuncture | 22 | 55% | 16 | 40% | 16 | 40% |
| Fingerstick | 9 | 23% | 12 | 30% | 18 | 45% |
| Both were the same / no preference | 9 | 23% | 12 | 30% | 6 | 15% |

These ratings were taken at the second lab session, after participants had experienced both kinds of draw.

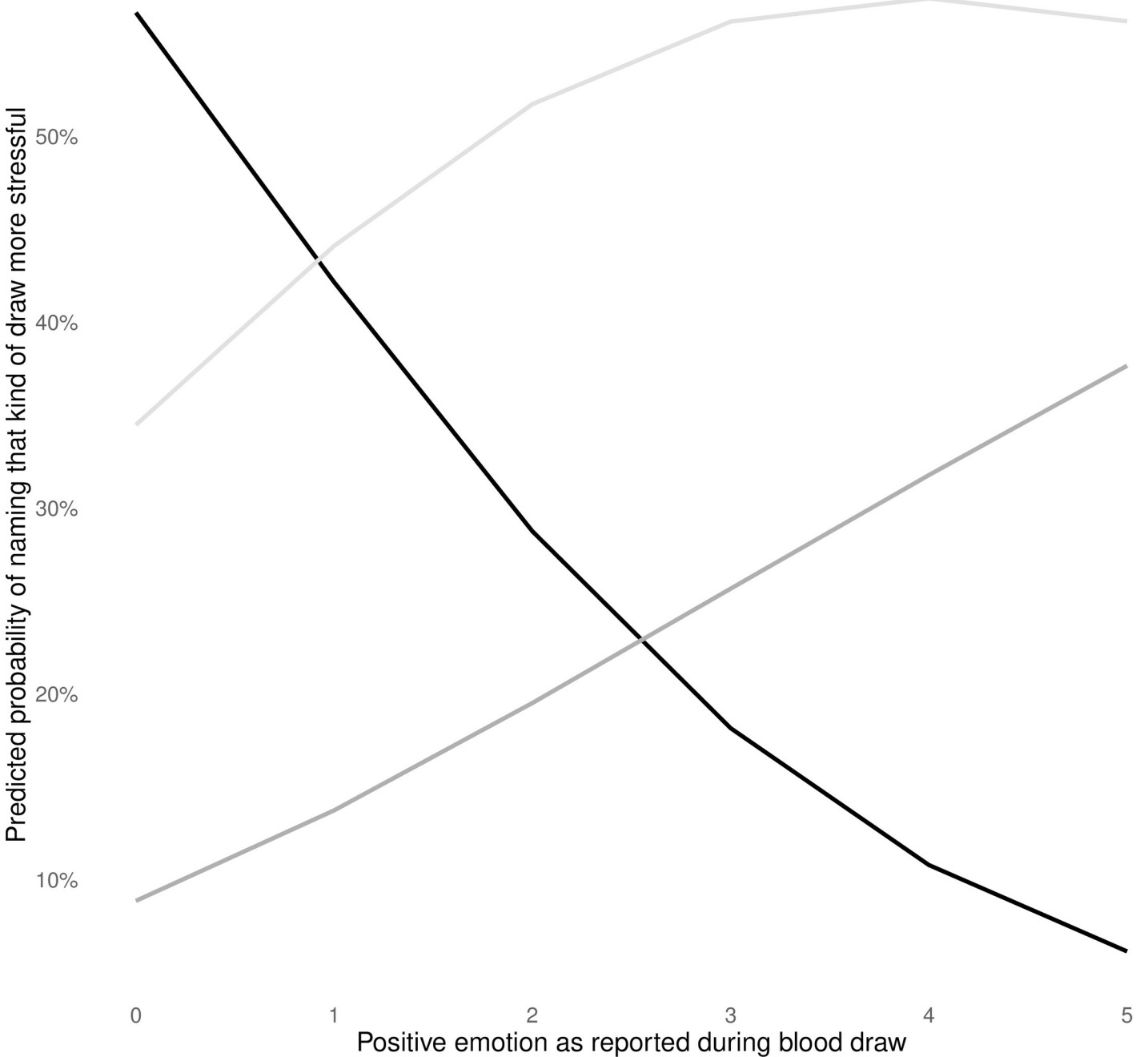

**Fig 5. Predicted probabilities for which type of draw was rated as more stressful, as predicted by positive emotion during blood draw procedures.**

**Predictors of preferences for future blood draws.**    Finally, in the best fit model for predicting participant's preferences for what kind of blood draw they would choose in the future, there were three predictors: negative emotion, self-reported physical stress, and the interaction

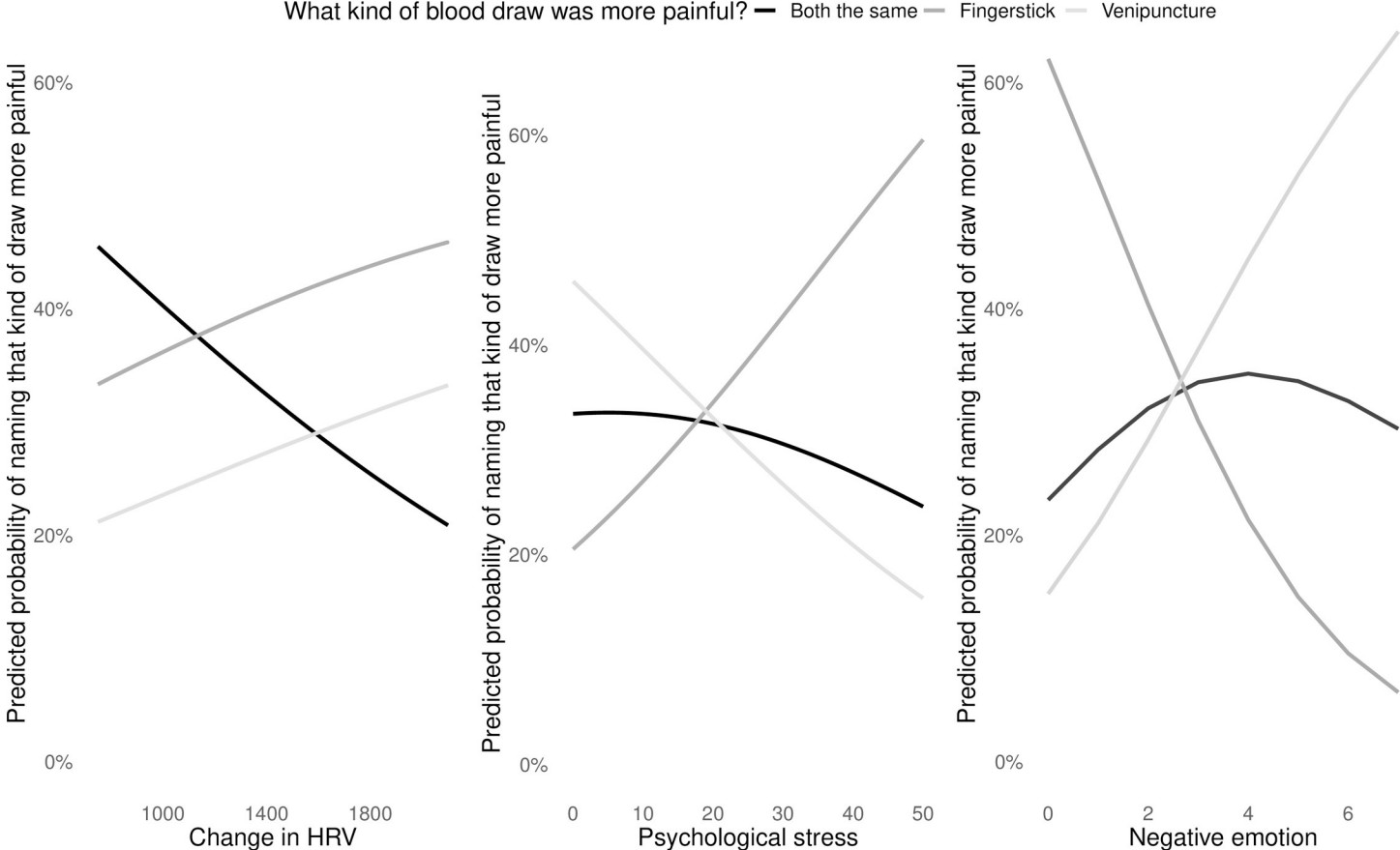

**Fig 6. Predicted probabilities for which type of blood draw was identified as more painful, as predicted by change in HRV, self-reported psychological stress, and negative emotion during blood draw procedures.** Change in HRV is represented as area under the curve with respect to ground (i.e., the integral of change over time); higher values indicate greater degree of change over time.

of positive emotion and draw type. The more negative emotion participants reported following blood draw, the more likely they were to prefer fingerstick. The more self-reported physical stress they reported during blood draw, the more likely they were to prefer venipuncture. The interaction between positive emotion and draw type was such that the higher a participant's positive emotion following venipuncture, the more likely they were to prefer fingerstick, while the higher their positive emotion following fingerstick, the more likely they were to prefer venipuncture (Fig 7).

## Discussion

The present study examined how the same individual responded to two different kinds of blood draw in terms of physical stress (cortisol, heart rate and HRV) and psychological stress (self-reported stress and emotion). Overall, the findings support the view that both fingerstick and venipuncture have moderate but consistently detectable effects on endocrine and autonomic activity that could potentially interfere with interpretation of results from experimental stress manipulations. Women's autonomic and psychological stress responses to either fingerstick or venipuncture had more similarities than difference; however, there may be important differences in endocrine response between draw types. This in turn implies that results from experiments using fingerstick vs. venipuncture may be comparable for some stress measures, but not others.

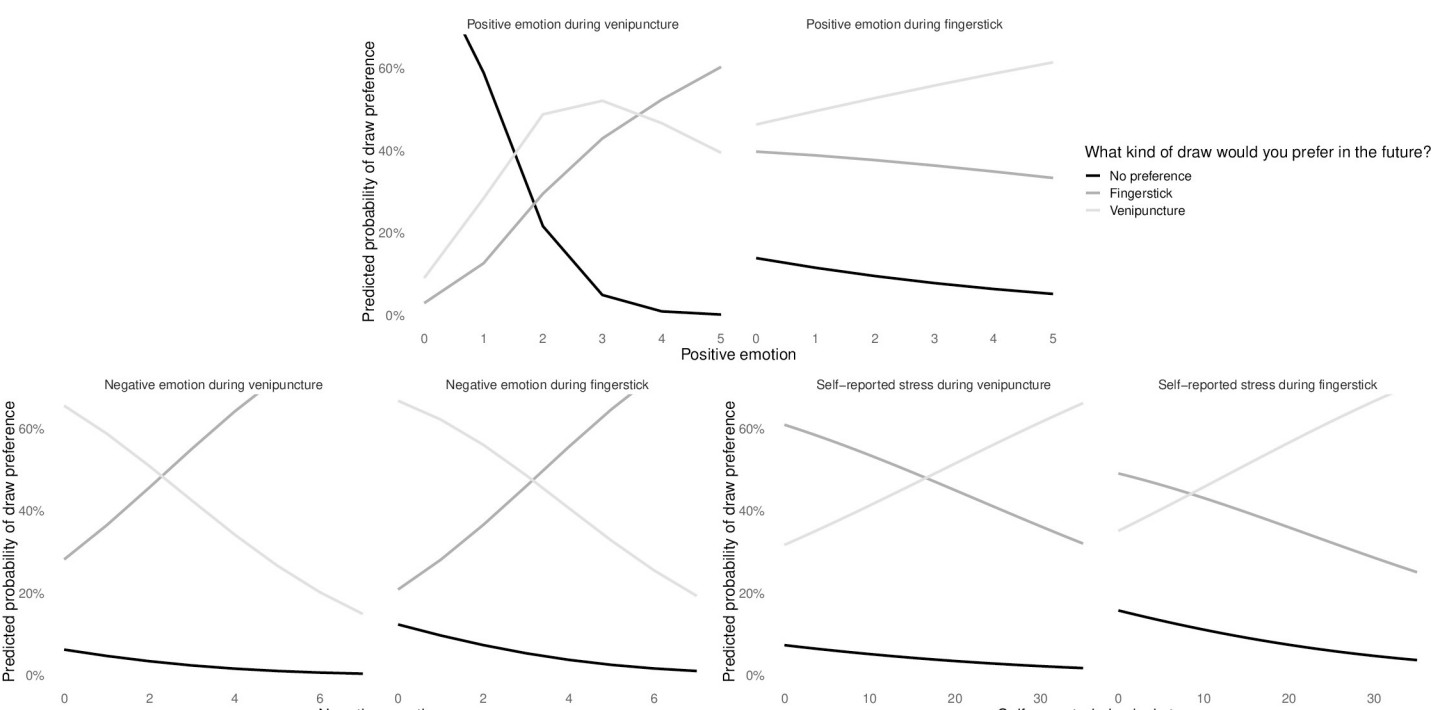

**Fig 7. Predicted probabilities for preferred draw type, as predicted by the interaction between draw type and positive emotion, negative emotion, and self-reported physical stress during blood draw procedures.**

For the most part, our pre-registered *a priori* hypotheses were unsupported: we did not find evidence that there were greater increases in physical stress during venipuncture than during fingerstick. In fact, we found quite the opposite: venipuncture elicited significant post-draw *decreases* in cortisol, while fingerstick remained relatively stable. And fingerstick and venipuncture elicited similar patterns of change in autonomic markers, with significant increase in parasympathetic activity during draw followed by return to baseline at recovery. In sum, we had not anticipated that blood draw would be associated with evidence of *decreased* stress responding. One possibility is that participants arrived at the lab in a state of anticipatory anxiety, knowing they would be undergoing blood draw. This interpretation is supported by the fact that on both days, participants pre-draw cortisol levels (average = 0.234 µg/dL) were on the high end of the normal range for afternoon samples in young women (typically, 0–0.359 µg/dL; [50]). If this is the case, then the decline in cortisol in venipuncture sessions could reflect recovery while the lack of decline in fingerstick sessions would reflect sustained physiological vigilance. It is not clear why fingerstick would be associated with such sustained stress, however, given participants did not generally report greater psychological stress or pain from fingerstick.

In terms of autonomic stress, participants showed significant decline in total heart rate from pre-draw to post-draw, which was sustained in the recovery period–again, supporting the idea of anticipatory anxiety. Pre-draw HRV (average = 43.1 ms) was on the low end of the normal range for healthy young women (32–93 ms; [51]). As lower HRV indicates higher sympathetic and/or lower parasympathetic activity, this too suggests that participants started the experimental sessions in a state of relatively elevated autonomic stress. However, HRV increased during the draw itself, and then returned to pre-draw levels in post-draw and recovery.

It seems unlikely that these data are indicating sudden relaxation during blood draw followed by return to an anxious state during recovery. It is more likely that changes in HRV during draw reflect a minor vasovagal reaction, followed by compensatory mechanisms to return autonomic balance to a resting state (as seen in the overall decline in total heart rate). That is, when the cardiovascular system detects threat of blood loss, it anticipates a possible drop in blood pressure and reacts by dilating blood vessels and slowing the heart: i.e., a vasovagal response [52]. The response is termed vaso*vagal* with good reason, as it is governed by sudden increases in vagal (parasympathetic) activity; our observed changes in HRV correspond to the rapid onset and equally rapid withdrawal of parasympathetic activity that are characteristic of a vasovagal response [53]. Vasovagal responses are particularly likely for individuals who have higher anticipatory anxiety prior to blood draw [54]. If these changes in HRV *do* reflect a vasovagal response, it would be a modest reaction, as no participant fainted or reported feelings consistent with pre-fainting (e.g., feeling lightheaded). And, it should be noted that women may be more susceptible to vasovagal reactions to venipuncture [55–58] (but see also [59]). As such, the findings presented here may reflect relatively larger changes in HRV to venipuncture than might be seen in a mixed sex/gender sample.

It is notable that within-person, there were equivalent HRV changes to venipuncture and fingerstick. Speculatively, individual differences in vasovagal reactivity to fingerstick could be used to predict the likelihood of similar reactions during venipuncture. Fingerstick tests for hemoglobin are typically done prior to blood donation, and could offer an opportunity to monitor changes in HRV that may detect donors at risk for syncope (fainting) during subsequent venipuncture. Our data suggest that one would need to observe changes in HRV specifically, not simply heart rate, to detect these subtle changes in autonomic balance: luckily, there is a growing number of commercially available mobile HRV monitors that have shown equivalent accuracy to ECG [60]. This could prove a particularly useful way to identify risk for adverse reactions in women who are first-time donors, who may be anxious simply due to lack of experience [61]; what is more, receiving feedback that a positive response to fingerstick would predict a similarly positive response to venipuncture may reassure these donors.

From a methodological standpoint, our finding that cortisol levels were relatively more stable following fingerstick would appear to highlight fingerstick as be a better choice for researchers looking to minimize the effects of blood draw on measurements of HPA activity. On the other hand, fingerstick did cause some small (albeit non-significant) fluctuations in cortisol levels, and both methods were associated with changes in autonomic activity, suggesting that a truly "stress-neutral" blood draw method is impossible. Our stress systems evolved to respond to potential threats to our physical integrity, and as such threats go, puncturing the skin and drawing blood is arguably the most fundamental. And, as noted above, it was likely that participants entered the session in a state of anticipatory anxiety. Taking all these considerations together, it seems most likely that the relative stability of cortisol during fingerstick does not reflect a true lack of change in HPA activity, but instead competing processes that canceled out their effects on the HPA end-product (i.e., cortisol). For example, if participants expected fingerstick to be less painful because the needle is smaller, the relief of their anticipatory anxiety (lowering stress) may have competed with pain being greater than anticipated (increasing stress). As such, these findings do call into question the assumption that fingerstick is necessarily less invasive than venipuncture.

That said, in both draws, the degree of change in physiological stress measures was moderate in absolute terms. For example, our largest observed effect, change in cortisol following venipuncture (~18% decline from pre-draw to recovery), was much smaller than the degree of change expected from normal diurnal variation in cortisol (in young women, an average of ~78% decline from morning to evening [50]. Nevertheless, researchers interested in subtle

effects may find even this moderate degree of change introduces an unacceptable level of potential bias in their measures of physical stress. And, it should be noted that there are important sex/gender differences in cortisol response, with men showing small but significantly greater increases in cortisol in response to pain than women [62, 63]. Thus, the findings presented here may reflect lower variability in cortisol response than would be expected in mixed sex/gender samples.

We did not find evidence for our *a priori* hypotheses that there would be greater psychological stress following venipuncture: rather, post-draw psychological stress, perceived physical stress, and emotional responses were similar across draw types. However, the interaction between physical and psychological stress measures did differ by draw type. Not surprisingly, participants whose cortisol fell more sharply across the session reported lower psychological stress regardless of type of draw. For fingerstick, this was reflected in higher perceived stress for participants whose cortisol rose in response to draw. But for venipuncture sessions, this meant that participants whose cortisol levels did not drop (i.e., were relatively more stable) reported more stress–even though their overall level of cortisol was lower. This suggests that it is the *relative* change in cortisol that contributed to perceived stress, rather than absolute level.

Change in HRV predicted participant's perceived physical stress, but not their subjective/psychological stress. Oddly, greater autonomic stress (i.e., lower HRV) was associated with *lower* perceived physical stress. This may simply be an anomaly. Because the definition of "physical stress" presented to participants specifically cued them to consider their heart rate (among other indicators such as sweating), participants may have been completing a pseudo-mindfulness exercise when rating their stress levels. This would be particularly relevant for participants with higher, and therefore more salient, heart rates. Indeed, some work has suggested that women who attend to their heart rate (and higher in anxiety sensitivity) are more accurate in detecting changes in heart rate to stress [64]. If these are not spurious findings, however, it is possible that blood draw is a unique stressor that may not have the expected associations between autonomic and psychological stress. Speculatively, if changes in HRV during blood draw do reflect vasovagal reactivity–that is, heightened parasympathetic activation in response to threat of blood loss–then perhaps subjective interpretation of the sensations accompanying parasympathetic activity would differ under this very specific set of conditions. In other words, although higher parasympathetic activity would generally indicate lower stress, when increasing parasympathetic activity is instead a signal of blood loss, it may generate higher psychological stress. Regardless, as this finding was very unexpected, it particularly warrants replication and caution in interpretation.

One prediction that was generally supported in our findings was that psychological measures were better predictors of perceived pain and preferences than physical stress markers. After having experienced both kinds of draw, participants were asked in a forced-choice question which method was more painful, which was more stressful, and which they preferred: post-draw emotional and psychological responses predicted all three, while only one physical stress measure (HRV) predicted participant choices, and only in one domain (pain). With one exception predictors were main effects, not interactions, indicating participant's psychological responses to either kind of draw were equally strong predictors of their ultimate preferences. For example, higher psychological stress–to either venipuncture *or* fingerstick–predicted greater likelihood of naming fingerstick as more painful. This suggests that participants' preferences for fingerstick or venipuncture may have more to do with their general attitudes towards getting blood drawn at all than any feature of the draw method itself, or their actual physical responses to those different methods. This parallels other findings that participants' presurgical ratings of hypervigilance and emotional bias uniquely predicts postsurgical pain above and beyond changes in cortisol reactivity [65, 66], as well as studies suggesting that

subjectively-rated anxiety sensitivity may predict participants self-reported pain and distress better than objectively measured autonomic stress response [67, 68].

This study had some significant strengths, including rigorous control of factors other than draw type, including comparing participants to themselves; measurement of multiple indices of physiological and psychological stress; and pre-registration of hypotheses and analytic plan. However, there were also some limitations worth noting. Because fingerstick can take slightly longer to complete than venipuncture for some participants (~1–3 minutes more), we were unable to completely control how much time the participant spent with the researcher in the room, which in turn may could have influenced participant's coping through additional distraction or social interaction. That said, this time difference is a feature of the method (fingers bleed more variably than veins), and likely to be similar in other research or clinical settings. This was a volunteer study, so unavoidably, people who were too distressed by the idea of blood draw would have self-selected out of participating. Of note, of our five dropouts, four were participants who completed fingerstick but no-showed for their venipuncture session: this suggests that even among participants who selected to be in a blood draw study, venipuncture was more daunting. Alternatively, completing the fingerstick session may have been sufficiently stressful for some participants that they did not want to return for the venipuncture session. Participants with no prior experience having their blood drawn were excluded, and thus these findings may not generalize to completely novice donors. And, because we did not ask how many times participants had previously had their blood drawn by either fingerstick or venipuncture, it is unknown if there were differences in the relative novelty of one procedure or the other. That said, the fact that participants rated both types of draw similarly stressful to other times they have had their blood drawn (such as at the doctors office) points to relatively minor differences in novelty effects.

Participants were young and physically healthy, and draws were completed in a research setting, so generalizability to older people or clinical settings is unknown. Similarly, participants were all female and predominantly White, and thus it is unclear the extent to which these patterns would hold people of different sexes/genders or racial/ethnic identities. As noted above, there are sex/gender differences in both autonomic and endocrine responses to stress and thus it would be reasonable to expect that these findings may not generalize to male-only or mixed sex/gender samples. Also, Black women face significant stressors in medical and research environments, including the stereotype that they are less susceptible to pain [69]; as such, it is reasonable to expect that they may develop very patterns of physical and psychological stress during medical procedures, and different coping strategies for dealing with that stress, than White women.

## Conclusions

We compared participant's autonomic, endocrine, and psychological stress responses to venipuncture vs. fingerstick. Contrary to predictions, fingerstick did not emerge as less stressful or invasive than venipuncture: in fact, where there were differences in stress response, these indicated recovery from anticipatory stress was faster for venipuncture. Psychological stress was related physical stress markers, but in unexpected ways: lower autonomic stress predicted higher self-reported stress, and relative decline, not absolute levels, of post-draw cortisol predicted self-reported stress. From a methodological perspective, these findings point to the need for caution when comparing data on stress reactivity in experiments that used different blood draw methods. These findings also suggest that while fingerstick may be "minimally invasive" in some ways, it is far from stress-neutral and as such, may interfere with interpretation of results from studies where stress response is a primary outcome.

## Author Contributions

**Conceptualization:** Tierney K. Lorenz.

**Data curation:** Tierney K. Lorenz.

**Formal analysis:** Tierney K. Lorenz.

**Funding acquisition:** Tierney K. Lorenz.

**Investigation:** Tierney K. Lorenz.

**Methodology:** Tierney K. Lorenz.

**Project administration:** Tierney K. Lorenz.

**Resources:** Tierney K. Lorenz.

**Software:** Tierney K. Lorenz.

**Supervision:** Tierney K. Lorenz.

**Validation:** Tierney K. Lorenz.

**Visualization:** Tierney K. Lorenz.

**Writing – original draft:** Tierney K. Lorenz.

**Writing – review & editing:** Tierney K. Lorenz.

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
