## [Decision Letter · Decision Letter 0]

24 Aug 2021

Autonomic, endocrine, and psychological stress responses to different forms of blood draw

PONE-D-21-14179

Dear Dr. Lorenz,

We’re pleased to inform you that your manuscript has been judged scientifically suitable for publication and will be formally accepted for publication once it meets all outstanding technical requirements.

Kind regards,

Ruud van den Bos, Ph.D.

Academic Editor

PLOS ONE

Journal Requirements:

1. Please provide additional details regarding participant consent. In the ethics statement in the Methods and online submission information, please ensure that you have specified what type you obtained (for instance, written or verbal, and if verbal, how it was documented and witnessed). If your study included minors, state whether you obtained consent from parents or guardians. If the need for consent was waived by the ethics committee, please include this information.

Reviewers' comments:

Reviewer's Responses to Questions

**Comments to the Author**

1. Is the manuscript technically sound, and do the data support the conclusions?

Reviewer #1: Yes

2. Has the statistical analysis been performed appropriately and rigorously? 

Reviewer #1: Yes

3. Have the authors made all data underlying the findings in their manuscript fully available?

Reviewer #1: Yes

4. Is the manuscript presented in an intelligible fashion and written in standard English?

Reviewer #1: Yes

5. Review Comments to the Author

Reviewer #1: In this interesting study blood drawn by fingestick is compared to venipuncture under rigorously controlled conditions. Autonomous and psychological stress markers were measured and gave additional information.

The authors investigated all kinds of correlations, which sometimes makes it hard to keep an overview. In the discussion, everything is well discussed and put into perspective.

6. PLOS authors have the option to publish the peer review history of their article (what does this mean?). If published, this will include your full peer review and any attached files.

Reviewer #1: No

---

## [Editor Report · Acceptance letter]

27 Aug 2021

PONE-D-21-14179 

Autonomic, endocrine, and psychological stress responses to different forms of blood draw 

Dear Dr. Lorenz:

I'm pleased to inform you that your manuscript has been deemed suitable for publication in PLOS ONE. Congratulations! Your manuscript is now with our production department. 

Kind regards, 

on behalf of

Dr. Ruud van den Bos 

Academic Editor

PLOS ONE